# Bacteriological quality and antimicrobial susceptibility profiles of isolates of ready-to-eat raw minced meat from hotels and restaurants in Arba Minch, Ethiopia

**Tomas Tonjo, Aseer Manilal**◯*, **Mohammed Seid***

Department of Medical Laboratory Science, College of Medicine and Health Sciences, Arba Minch University, Arba Minch, Ethiopia

* aseermanilal@gmail.com (AM); mohamedseid2005@gmail.com (MS)

**Data Availability Statement:** All relevant data are within the paper.

**Funding:** The author(s) received no specific funding for this work.

## Abstract

In Ethiopia, the bacteriological quality of ready-to-eat raw meat is of a great public health concern as it can serve as a source of meat-borne pathogens and worsen the transmission of antimicrobial resistant bacteria, and hence this cross-sectional study, done on 257 meat samples (ie., 169 beef, 50 mutton and 38 chevon) from randomly selected hotels and restaurants (n = 52). Approximately 25 gm of meat samples were taken bi-weekly and subjected to quantitative and qualitative analyses; antimicrobial susceptibility tests were done as per the Kirby-Bauer disk diffusion method. It was found that 13.2 (n = 34), 17.5 (n = 45) and 21.8% (n = 56) samples exceeded the permissible limit for total viable and coliform and *S. aureus* counts, respectively. At the same time, 24.9% (n = 64) surpassed the bacteriological limit permissible for consumption. Overall, 36.6% (n = 94) of samples were extrapolated as unsatisfactory for consumption due to high bacterial load and or the presence of pathogens. Five different bacterial spp. such as *E. coli* 65% (n = 167), *S. aureus* 59% (n = 152), *Salmonella* spp. 28.4% (n = 73), *Campylobacter* spp. 14.4% (n = 37) and *Shigella* spp. 4.3% (n = 11) were isolated in varied proportions. Alarmingly, 60% (n = 264) of the isolates were multi-drug resistant and 51% of *S. aureus* were found to be MRSA.

## Introduction

The World Health Organization (WHO) defines foodborne diseases (FBDs) as "ailments of infectious or toxic nature caused by or suspected to be caused by the consumption of food or water" [1]. It is estimated that the continent of Africa had the highest FBDs burden in 2010, with over 91 million falling ill and 137,000 deaths annually which represent one-third of the total global deaths. Ethiopia ranks second after Nigeria in this regard and FBDs pose serious threats to the health of people in the country [2, 3].

Diarrheal diseases due to foodborne pathogens are responsible for 70% of the associated death toll worldwide [4]. In Ethiopia, mortality due to diarrheal diseases was 2.6 million in 2010, and they are the second leading cause of premature death, falling just behind the toll due

**Competing interests:** The authors have declared that no competing interests exist.

to lower respiratory tract infections [5]. The most common bacterial pathogens which cause diarrheal diseases worldwide are enteric, particularly non-typhoidal *Salmonella* spp., *Escherichia coli* (entero-pathogenic, entero-toxigenic and entero- hemorrhagic), *Campylobacter* spp., *Shigella* spp. and *Staphylococcus aureus* [2, 5].

Meat is one of the food items that are rich in the nutrient matrix which provides a suitable environment for the proliferation of spoilage microorganisms and common foodborne pathogens. The most common source of bacterial diarrheal diseases caused by meat chains are of animal origin and are connected to the environment, meat handlers/processors and the processing equipment. Even though the source of bacteria varies, raw meat is confirmed as a common vehicles of foodborne diseases [6, 7]. The extent of microbial contamination and composition reflects the quality of meat. Moreover, safety problems that meat consumers face most often in hotels and restaurants are related to microorganisms, particularly bacterial pathogens [7].

Above all, the most staggering and challenging situation is that contaminated meat can carry antimicrobial-resistant pathogens and can seriously endanger human health. Predominantly, raw meat consumers are highly vulnerable and resistant genes can be transferred to the normal microbiota existing in their gastrointestinal tract [8]. Transmission of these resistant bacteria to humans via meat is evident as animals such as cattle, sheep and goats are important reservoirs of *E. coli*, *Campylobacter* spp., *S. aureus* and *Salmonella* spp. [9]. Infections caused by resistant bacteria cause more severe ailments and often require very expensive treatments with higher risks of side effects and have been recognized by WHO as the most vital health issue of the 21$^{st}$ century [10]. The problem is even more alarming in developing countries, where there exists an enormous burden of infectious diseases, accompanied by lack of surveillance networks, paucity of testing laboratories, and inadequate diagnostics [11].

In Ethiopia, raw beef, chevon and mutton are traditional delicacies and consumer demand is increasing due to their high nutritional value. Although healthy meat from beef cattle, sheep and goats provide nutritionally beneficial components, illegal slaughtering in open fields, unhygienic slaughter practices in abattoirs and unscientific processing in food establishments contaminate them with pathogenic microorganisms. These harmful pathogens create several diseases in humans leading to increased morbidity, mortality and high cost of treatment. The general masses comprising all age groups in the study area, Arba Minch town, have the habit of consuming ready-to-eat (RTE) raw meat in various delicacies. Nevertheless, many hotels and restaurants are also serving RTE raw meats, without assessing their bacteriological quality, mainly due to the lack of maintaining the quality by the food safety and inspection department in the town. A careful literature survey indicates that studies in this context are sorely lacking. Therefore, the present study was initiated to evaluate the bacteriological quality, types of common bacterial isolates and their antimicrobial susceptibility profiles in RTE minced meat (beef, mutton and chevon) served in selected hotels and restaurants of Arba Minch town.

## Materials and methods

### Study area and period

A cross-sectional study was carried out over a period of six months (July 1, 2020, to December 31, 2020) in the town of Arba Minch, Gamo Zone, southern Ethiopia. The town has only one slaughterhouse and the number of animals slaughtered per day varied from time to time, for instance, during fasting, festival and non-fasting periods. The average number of cattle slaughtered per month approximates 600 excluding sheep and goats. There are around 60 hotels and restaurants in the town itself and they serve RTE minced beef, mutton and chevon. Ethical approval for this study was obtained from the Institutional Review Board of College of

Medicine and Health Science, Arba Minch University, Arba Minch, Ethiopia (Ref. No IRB/ 174/12/17/03/2020).

## Sample size determination

**Meat samples.** A risk assessment guideline jointly set by the Food and Agriculture Organization and the WHO was used to fix the microbiological criteria for meat samples [12]. The analysis unit of 25gm meat was collected in the mid-morning (9 to 11 AM) from 52 hotels and restaurants once in every two weeks over a period of two and a half months, starting from 01 July to the 15 September. A total of 260 samples comprising 170 beef, 50 mutton and 40 chevon were randomly collected. One beef sample and two samples of chevon were rejected due to sampling error, and thus finally 257 RTE minced meat samples were subjected to bacteriological analysis.

**Sampling technique.** The number of hotels and restaurants serving mutton, chevon and beef are 10, 8 and 34 (total 52) as per the proportional allocation, respectively. Finally, one sample per hotel and restaurant, once in two weeks over a period of two and a half months was randomly collected. The inclusion criterion is RTE raw minced meat (regional name of RTE: Kurt/Kitfo) that is processed for direct consumption, whereas that of the exclusion was minced meat which is processed for cooking.

**Sample collection.** Samples were carefully collected by using sterile tweezers to ensure the avoidance of extraneous contamination and were placed in a plastic bag and kept in an ice box and were immediately transported to the Medical Microbiology and Parasitology Laboratory, Department of Medical Laboratory Science, for bacteriological analysis, within 2 h of collection [13].

## Bacteriological analyses

**Quantitative bacteriological analysis.** Twenty-five grams of meat immersed in 225 ml of sterile water was blended for 2 minutes. Serial dilutions (up to $10^{-6}$) were performed to quantify the microbial analyses such as total viable count (TVC), total coliform count (TCC) and total *S. aureus* count (TSC) [14].

**Total viable count.** About 0.1ml, from each serial dilution, was spread onto the plate count agar (Hi media, India) and incubated at 37˚C for 24 h. After incubation, plates showing 30 to 300 colonies were counted and the results were calculated according to ISO 4833: 1–2013 [14].

Finally, the results were classified according to the standard criteria set by hazard analysis and risk assessment (for the management of food safety and quality) for bacteriological limits permissible for human consumption [15].

**Total coliform and *S. aureus* count.** About 0.1ml from each dilution was spread onto the media such as MaConkey agar and mannitol salt agar (Hi media, India), for the total coliform count and *S. aureus* count, respectively; incubated at 37˚C for 24 h [16, 17]. In the case of TCC, results were rated according to the standard criteria set by the hazard analysis and risk assessment for bacteriological limits permissible for consumption [15].

**Qualitative bacteriological analyses of samples.** Suspected colonies of *S. aureus*, *E. coli*, *Salmonella* spp., *Shigella* spp., and *Campylobacter* spp., were identified as per the conventional techniques described elsewhere [16–20].

## Antimicrobial susceptibility testing

The antimicrobial susceptibility test was performed on Mueller Hinton agar (Hi media, India) by using the Kirby-Bauer disk diffusion method according to the criteria set by CLSI, 2016,

2019 [21, 22]. For *Campylobacter* spp., Mueller Hinton agar supplemented with 5% sheep blood was used and plates were incubated at 42˚C for 18–24 h in a candle jar. Diameters of zones of inhibition around the disks were measured using a ruler and categorized as susceptible, intermediate and resistant according to the standard table described in CLSI. The multi-drug resistance (MDR) in this study was extrapolated as resistance to three or more classes of antibiotics tested [23].

The antibiotic disks used for various bacteria included the following; penicillin (P-10 units), ampicillin (AM-10 μg), amoxicillin/clavulanic acid (AUG-20/10μg), erythromycin (E-15 μg), tetracycline (TE-30μg), ciprofloxacin (CIP-5μg), co-trimoxazole (TMP-SMX-1.23/23.75μg), gentamicin (GM-10μg), chloramphenicol (C-30μg), clindamycin (CL-2μg), meropenem (M-10 μg) and cefoxitin (CXT-30μg) [22].

**Phenotypic detection of methicillin resistance of *S. aureus* (MRSA).**   Isolates of *S. aureus* were tested to detect methicillin resistance according to the CLSI guidelines by using a cefoxitin disk and a zone of inhibition ≤21 mm was confirmed as corresponding to MRSA [22].

## Quality control

The quality of data was ensured from collection onwards to the final laboratory identification by following the standard operating procedure (in-house SOP). The performance of prepared media was checked by inoculating the control (reference) strains such as *E. coli* ATCC 25922, *C. jejuni* ATCC 700819, *S. flexineri* ATCC 12022, *Salmonella* Typhimurium ATCC 14028 and *S. aurues* ATCC 25923 which were obtained from Ethiopian Public Health Institute.

## Data analysis

Data were analyzed using SPSS, Chicago, IL, the USA, Windows, version 25. Results of bacterial counts were expressed in terms of mean log10 and were compared with standards. The isolation rate of common bacterial isolates, their prevalence and antimicrobial susceptibility were expressed in percentages. The mean microbial counts of TVC, TCC and TSC among different types of meat were compared by the analysis of variance; $p$ value < 0.05 was considered statistically significant.

## Results

### Quantitative analyses (TVC, TCC and TSC)

Of the total raw minced meat samples analyzed, 55.3% (n = 142) had TVC, 56.8% (n = 146) had TCC and 46.7% (n = 120) had TSC well below the permissible limits with respect to the microbial load, making them satisfactory and hence safe for consumption (Table 1). Concurrently, 31.5% (n = 81) of TVC, 25.7% (n = 66) of TCC and 31.5% (n = 81) of TSC of meat samples had shown only marginal bacteriological load indicating that consumption was allowed; but not fully satisfactory, however, only having intermediate status. On the contrary, 13.2% (n = 34), 17.5% (n = 45) and 17.5% (n = 45) of meat samples had unacceptable ranges of bacteriological load with regard to TVC, TCC and TSC respectively. The most alarming factor is that 10.5% (n = 4) of chevon, 8% (n = 4) of mutton, and 1.8% (n = 3) of beef samples were contaminated with *S. aureus* and their TSC loads remained in a potentially harmful range. Based on the extent of bacteriological loads, the overall percentage of meat samples which was unsatisfactory for consumption was extrapolated as 25% (n = 64). However, a total of 42.8 (n = 110) and 32.3% (n = 83) of meat samples were found to be satisfactory and marginally satisfactory respectively; they were acceptable for consumption as far as the permissible limits of microbiological standards are concerned.

In the present study, even the detection of a single colony of *Salmonella* spp., *Campylobacter* spp. and or *Shigella* spp. in the meat samples was regarded as unacceptable for consumption in

**Table 1. Bacteriological load of different types of RTE raw minced meats served in hotels and restaurants of Arba Minch town from 01 July to 31 December, 2020.**

| Category | Meat samples (n) | Bacteriological load based on TVC, TCC and TSC | | | | | | Total extrapolation of load n (%) |
|---|---|---|---|---|---|---|---|---|
| | | TVC | Log of mean TVC | TCC | Log of mean TCC | TSAC | Log of mean TSC | |
| *Satisfactory | | 142(55.3) | | 146 (56.8) | | 120 (46.7) | | |
| | Beef (n = 169) | 104 (61.54) | 2.7 | 111 (65.7) | 0.9 | 92(54.4) | 0.9 | 110(42.8) |
| | Mutton (n = 50) | 22 (44) | 3.06 | 20 (40) | 1.02 | 19(38) | 1.28 | |
| | Chevon (n = 38) | 16 (42.1) | 3.05 | 15 (39.5) | 1.37 | 9(23.7) | 1.32 | |
| **Marginal | | 81(31.5) | | 66(25.7) | | 81(31.5) | | |
| | Beef (n = 169) | 46 (27.22) | 6.13 | 36 (21.3) | 2.19 | 48(28.4) | 2.68 | 83(32.3) |
| | Mutton (n = 50) | 19 (38) | 6.39 | 17 (34) | 2.34 | 17(34) | 3.01 | |
| | Chevon (n = 38) | 16 (42.1) | 6.25 | 13 (34.2) | 2.29 | 16(42.1) | 2.98 | |
| ***Unsatisfactory[a*] | | 34(13.2) | | 45(17.5) | | 45(17.5) | | |
| | Beef (n = 169) | 19 (11.24) | 7.03 | 22 (13) | 3.01 | 26(15.4) | 4.21 | 53(20.6) |
| | Mutton (n = 50) | 9(18) | 7.50 | 13 (26) | 3.33 | 10(20) | 4.43 | |
| | Chevon (n = 38) | 6 (15.8) | 7.61 | 10 (26.3) | 3.71 | 9(23.7) | 4.29 | |
| ****Potentially harmful[a*] | | ND | ND | ND | ND | 11(4.3) | | |
| | Beef (n = 169) | ND | ND | ND | ND | 3(1.8) | 4.97 | 11(4.3) |
| | Mutton (n = 50) | ND | ND | ND | ND | 4(8) | 5.03 | |
| | Chevon (n = 38) | ND | ND | ND | ND | 4(10.5) | 5.03 | |

[a*] Meat samples unfit for human consumption due to high microbial load is extrapolated as 24.9% (64 = 53 +11) based on the microbial load

*satisfactory: TVC ≤ 5.7log, TCC ≤ 1.7log and TSC ≤ 2log10

**marginal: 5.7log < TVC ≤ 6.7 log, 1.7 log < TCC ≤ 2.7 log and 2 log < TSC ≤ 3.7 log.

***Unsatisfactory: 6.7 log < TVC < 8.7 log, 2.7 log < TCC < 4.7 log and 3.7 log < TSC <4.7 log

****potentially harmful: TVC ≥ 8.7 log, TCC ≥ 4.7 log and TSC ≥ 4.7log.

ND: Not detected.

the raw state. Thus, altogether 36.6% (n = 94) of samples were found to be unfit for consumption due to higher microbial load and or the presence of pathogenic microorganisms.

## Comparative analyses of the log of mean TVC, TCC and TSC

There exist statistically significant differences among the mean counts of samples of beef and mutton and also while comparing the samples of beef and chevon ($p$-value <0.05), with respect to TVC, TCC and TSC (Table 2); however, no statistically significant differences ($p$-value > 0.05) were found among the mean counts of mutton and chevon samples.

## Qualitative bacteriological analyses

As per the bacteriological analysis, totally 440 isolates belonging to five different genera were identified. Isolates of *E. coli* were the most frequently detected, 65% (n = 167), followed by *S.*

**Table 2. The log of total mean (±SE) of TVC, TCC and TSC of different meat sources.**

| Meat samples | Log of mean TVC | Log of mean TCC | Log of mean TSC |
|---|---|---|---|
| Beef (n = 169) | 4.19 ± 0.23 | 1.23 ±0.11 | 1.90 ±0.12 |
| Mutton (n = 50) | 5.15 ± 0.47 | 2.09±0.27 | 2.91 ±0.23 |
| Chevon (n = 38) | 5.14 ± 0.44 | 2.11±0.25 | 3.00 ±0.25 |

SE: standard error

**Table 3. Bacterial isolates from different RTE raw minced meats served in hotels and restaurants of Arba Minch town from 01 July to 31 December, 2020.**

| Bacterial isolates | Type of meat | Percentage of bacterial isolates n (%) | Overall prevalence of bacterial isolates n (%) |
|---|---|---|---|
| *E. coli* | Beef | 110(65) | 167(65) |
| | Mutton | 30(60) | |
| | Chevon | 27(71) | |
| *S. aureus* | Beef | 94(55.6) | 152(59) |
| | Mutton | 30(60) | |
| | Chevon | 28(73.7) | |
| *Salmonella* spp. | Beef | 45(26.6) | 73(28.4) |
| | Mutton | 17(34) | |
| | Chevon | 11(28.9) | |
| *Campylobacter* spp. | Beef | 20(11.8) | 37(14.4) |
| | Mutton | 9(18) | |
| | Chevon | 8(21) | |
| *Shigella* spp. | Beef | 8(5.7) | 11(4.3) |
| | Mutton | 1(2) | |
| | Chevon | 2(5.3) | |

*aureus*, 59% (n = 152), *Salmonella* spp., 28.4% (n = 73), *Campylobacter* spp., 14.4% (n = 37) and *Shigella* spp., 4.3% (n = 11) (Table 3).

## Antimicrobial susceptibility profiles

Isolates of bacteria belonging to five different genera showed broad variations in their resistance/susceptibility profiles. Of the 73 isolates of *Salmonella* spp. tested against six antibiotics, relatively lower resistance was only observed against chloramphenicol (35%), augmentin (34.2%) and tetracycline (30%). With regard to the susceptibility profile, 98.6% of isolates were found to be susceptible to meropenem followed by co-trimoxazole (92%), and ampicillin (75.3%). Out of the 37 isolates of *Campylobacter* spp. tested against a set of nine antibiotics, moderate levels of resistance were shown only against two antibiotics such as augmentin (54.5%) and ampicillin (54%). As a matter of fact, the majority of isolates showed susceptibility toward antibiotics such as ciprofloxacin (95%), co-trimoxazole (89%) and erythromycin (87%). Of the eleven isolates of *Shigella* spp. tested against seven antibiotics, the highest degree of resistance was found against ampicillin (90%). Invariably, all isolates (100%) were susceptible to meropenem and co-trimoxazole whereas 90% were susceptible to ciprofloxacin. Out of the 152 isolates of *S. aureus* tested, 90 and 62% showed resistance to penicillin and erythromycin respectively. It is to be noted that the majority of the isolates were susceptible to co-trimoxazole (98%), chloramphenicol (97%), clindamycin (96%) and ciprofloxacin (95.4%). In the case of *E. coli* isolates, a fairly high degree of resistance was noticed against ampicillin (60%). However, the majority of them were found to be susceptible to meropenem (89%), augmentin (84%) and tetracycline (79%). Invariably, all of them showed extreme susceptibility (100%) to ciprofloxacin, co-trimoxazole and chloramphenicol (Table 4).

## Multi-drug resistant bacterial isolates

The most alarming result obtained from our study is that 60% (n = 264) of isolates were MDR (Table 5). Higher percentage of MDR isolates were detected in mutton, ie., 67.8% (n = 59) followed by beef, 58.8% (n = 163) and chevon 55.3% (n = 42). Among the bacterial isolates, 65.7% (n = 48) of *Salmonella* spp., 56.8% (n = 21) of *Campylobacter* spp., 72.7% (n = 8) of *Shigella* spp., 59.9% (n = 91) of *S. aureus* and 63.5% (n = 106) of *E. coli* were found to be MDR

**Table 4. The antimicrobial susceptibility profiles of bacterial isolates from RTE raw minced meats served in hotels and restaurants of Arba Minch town from 01 July to 31 December, 2020.**

| Antibiotics | Antimicrobial susceptibility profiles n (%) | | | | | | | | | | | | | | |
|---|---|---|---|---|---|---|---|---|---|---|---|---|---|---|---|
| | Gram-negative bacterial isolates | | | | | | | | | | | | Gram-positive bacterial isolates | | |
| | *Salmonella* spp. (n = 73) % | | | *Shigella* spp. (n = 11) % | | | *Campylobacter* spp. (n = 37) % | | | *E. coli* (n = 167) % | | | *S. aureus* (n = 152) % | | |
| | S | I | R | S | I | R | S | I | R | S | I | R | S | I | R |
| Penicillin | NT | NT | NT | NT | NT | NT | NT | NT | NT | NT | NT | NT | 15(10) | NT | 137(90) |
| Ampicillin | 55(75.4) | 5(6.8) | 13(17.8) | 1(9.1) | 0 | 10(90.9) | 12(32.4) | 5(13.5) | 20(54.1) | 50(30) | 17(10) | 100(60) | NT | NT | NT |
| Augmentin | 28(38.4) | 20(27.4) | 25(34.2) | 5(45.5) | 1(9) | 5(45.5) | 15(40.5) | 2(5.4) | 20(54.1) | 140(84) | 7(4) | 20(12) | 84(55) | 12(8) | 56(37) |
| Cefoxitin | NT | NT | NT | NT | NT | NT | NT | NT | NT | 14(8) | 100(60) | 53(32) | 74(48.7) | NT | 78(51.3) |
| Erythromycin | NT | NT | NT | NT | NT | NT | 32(87) | 0 | 5(13) | NT | NT | NT | 44(29) | 14(9) | 94(62) |
| Tetracycline | 47(65) | 4(5) | 22(30) | 9(82) | 1(9) | 1(9) | 22(60) | 2(5) | 13(35) | 132(79) | 13(8) | 22(13) | 68(45) | 34(22) | 50(33) |
| Ciprofloxacin | NT | NT | NT | 10(90.9) | 0 | 1(9.1) | 35(95) | 0 | 2(5) | 167(100) | 0 | 0 | 145(95.4) | 5(3.3) | 2(1.3) |
| Co-trimoxazole | 67(92) | 2(3) | 4(5) | 11(100) | 0 | 0 | 33(89) | 3(8) | 1(3) | 167(100) | 0 | 0 | 149(98) | 3(2) | 0 |
| Gentamicin | NT | NT | NT | NT | NT | NT | 24(64.9) | 5(13.5) | 8(21.6) | 162(97) | 0 | 5(3) | 83(55) | 8(5) | 61(40) |
| Chloramphenicol | 37(51) | 10(14) | 26(35) | 5(45.5) | 1(9) | 5(45.5) | 22(59.5) | 7(18.9) | 8(21.6) | 167(100) | 0 | 0 | 147(97) | 0 | 5(3) |
| Meropenem | 72(98.6) | 1(1.4) | 0 | 11(100) | 0 | 0 | 21(56.8) | 4(10.8) | 12(32.4) | 149(89) | 10(6) | 8(5) | NT | NT | NT |
| Clindamycin | NT | NT | NT | NT | NT | NT | NT | NT | NT | NT | NT | NT | 146(96) | 0 | 6(4) |

NT: indicating the antibiotics discs that were not tested against pathogens

(Table 5). More than 51% of isolates of *S. aureus* showed a zone of inhibition ≤21 mm in cefoxitin disk diffusion assay and were extrapolated as methicillin-resistant *S. aureus* and all of them were also found to be MDR.

## Discussions

In the present study, only 42.4% (n = 109) of RTE meat samples were satisfactory for consumption and 36.6% (n = 94) were found to be unsatisfactory. The widespread practice of consuming raw meat can be considered to be a major risk of foodborne infections in Ethiopia [24].

Classifications pertaining to the combined results of quantitative analysis, in terms of TVC, TCC and TSC and the presence of bacterial pathogens, in RTE meats are limited, and the details cannot be found in the open literature so often. On analyzing the data obtained, appreciable differences ($p < 0.05$) were found in the mean count of TVC of beef samples compared to those of mutton and chevon. However, only marginal differences ($p < 0.05$) were observed in the mean TVC of mutton and chevon samples. The TCC and TSC of mutton and chevon samples, were statistically identical, which could be linked to the similarity in hygiene practices adopted during processing. The mean counts of TVC, TCC and TSC in beef were 4.19, 1.23 and 1.9, respectively. These values are much lower than that found in similar studies conducted in Addis Ababa (8.34, 4.69 and 5.36), [25] and Adama (5.2, 1.72 and 5.74), Ethiopia [26], Nigeria (4.53, 3.97 and 3.88) [27], Ivory Coast (8.1, 4.73 and 2.43) [28], Ghana (TVC ranged between 3.34 and 4.1 and TCC ranged between 2.28 and 2.87) [29] and South Africa (TVC ranged between 2.51 and 4.32 and TCC ranged between 2.58 and 3.91) [30].

The mean counts of bacterial load in chevon samples (ie., TVC 5.14 and TCC 2.11) observed in our study are lower than the values obtained in a previous study conducted in Nepal (ie., TVC 7.92 and TCC 6.37) [31]; however, the mean count of TVC and TCC was comparable to the values of a couple of studies conducted in Bangladesh (TVC 5.24±0.42 and TCC 2.63±0.09) [32] and Nigeria in the case of TVC (5.4) [33]. The mean counts of TVC, TCC and

**Table 5. MDR profiles of bacterial isolates from RTE raw minced meats served in hotels and restaurants of Arba Minch town from 01 July to 31 December, 2020.**

| MDR | Bacterial isolates | Beef n = 277(%) | Mutton n = 87 (%) | Chevon n = 76 (%) |
|---|---|---|---|---|
| *R3 | Salmonella spp. (n = 73) | n = 45 | n = 17 | n = 11 |
| | | 19(42.2) | 5(29.4) | 2(18.2) |
| | Campylobacter spp. (n = 37) | n = 20 | n = 9 | n = 8 |
| | | 5(25) | 2(22.2) | 3(37.5) |
| | Shigella spp. (n = 11) | n = 8 | n = 1 | n = 2 |
| | | 2(25) | | 1(50) |
| | S. aureus (n = 152) | n = 94 | n = 30 | n = 28 |
| | | 38(40.4) | 12(40) | 7(25) |
| | E. coli (n = 167) | n = 110 | n = 30 | n = 27 |
| | | 31(28.2) | 10(33.3) | 11(40.7) |
| *R4 and above | Salmonella spp. (n = 73) | n = 45 | n = 17 | n = 11 |
| | | 10(22.2) | 7(41.2) | 5(45.5) |
| | Campylobacter spp. (n = 37) | n = 20 | n = 9 | n = 8 |
| | | 6(30) | 3(33.3) | 2(25) |
| | Shigella spp. (n = 11) | n = 8 | n = 1 | n = 2 |
| | | 3(37.5) | 1(100) | 1(50) |
| | S. aureus (n = 152) | n = 94 | n = 30 | n = 28 |
| | | 18(19) | 6(20) | 10(35.7) |
| | E. coli (n = 167) | n = 110 | n = 30 | n = 27 |
| | | 31(28.2) | 13(43.3) | 10(37) |
| Total % (n) | | 163(58.8) | 59(67.8) | 42(55.3) |
| Cumulative total n (%) | | 264(60) | | |

*R3, *R4: isolates respectively resistant to three and four antibiotics from different classes.

TSC in mutton were found to be 5.15, 2.09 and 2.91, respectively. Our results are by and large compared to the values of TVC (4.72 for chevon and 4.39 for mutton) reported in a study conducted in Ghana [34] and South Africa (TVC of mutton ranged from 2.48 to 4.38 and TCC of mutton was between 2.48 and 3.45) [30].

Even though the overall mean counts from our study remained lower than that found in other similar studies conducted in different regions of Ethiopia, we observed the presence of pathogens like *Salmonella* spp., *Campylobacter* spp. and *Shigella* spp. This kind of contamination of raw meats could pose serious public health problems. In our study, the rate of isolation of *Salmonella* spp. was found to be 28.4%, and was the highest in mutton samples (34%) followed by chevon (28.9%) and beef (26.6%). The percentage isolation rates of *Salmonella* spp. from mutton and chevon samples were higher than the values corresponding to other studies conducted in different parts of Ethiopia [35, 36]. Similarly, beef samples also had shown higher isolation rates compared to a study conducted in Wolaita Sodo [36]; nevertheless, remained lower than the values reported by other studies conducted in Namibia [37] and South Africa [30]. Finally, values obtained in the current study were also higher than the pooled estimates of contaminated minced beef (8.34%) and mutton (11.86%) reported earlier in Ethiopia [38]. The possible reasons for these differences in the isolation rates of *Salmonella* spp. could be the fluctuations in slaughtering practices, post-slaughter handling procedures and the standards of general hygiene maintained at various stages of the processing chain [39]. Moreover, a previous work done in Arba Minch reported that the isolation rate of Salmonella sp. from food handlers was 6.9% [40].

The overall isolation rate of *Campylobacter* spp. is found to be 14.4%, corresponding to 11.8, 18 and 21% in the case of beef, mutton and chevon samples respectively; however, the rate of isolation from beef samples in the current study was higher than the results of a previous investigation reported from Addis Ababa, (6.5%) [41], but is lower than the outcome of a study done in Nigeria (12.9%) [42]. The isolation rate of *Campylobacter* spp. from mutton was lower than that found in a previous study conducted in Debre Berhan, Ethiopia (21.4%) [43]. Nevertheless, it is slightly higher than the values found in a past study reported from Addis Ababa (10.5%) [41]; the isolation rate of *Campylobacter* spp. in chevon was higher than that reported in a study cited above (7.6%) [41]. The higher level of isolation rates observed could be a reflection of the contamination of carcasses with the animals' intestinal contents during manual skinning, evisceration, washing and further processing in slaughterhouses or can be due to more frequent contact occurring between the hands of operators and their knives [44]. A recent meta-analysis revealed that the pooled prevalence of *Campylobacter* sp. in Ethiopia was 10.2% with a higher prevalence in animals [45].

*Shigella* spp. is considered as a foodborne pathogen and they originate from the environment including water [33]. The isolation rate of *Shigella* spp. (4.3%) was found to be much higher than that obtained from a previous study done in Jimma, Ethiopia (0.6%) [46]. However, it is lower than the result of studies reported from Gondar, Ethiopia 10.5% [47] and also Nepal (6%) [31]. In addition, a previous work done in Arba Minch, Ethiopia reported that the isolation rate of *Shigella* sp. among food handlers was 3% [40]. According to WHO, 25% of diarrhoea is caused by food contaminated with *E. coli* [1]. The most frequently isolated bacteria was *E. coli* (65%), ie., 65, 60 and 71% from beef, mutton and chevon, respectively. Chevon samples used in this study were highly contaminated with *E. coli* (71%) and the extent was higher than that reported in Nepal (46.7%) [31]. In the case of beef, the isolation rate of *E. coli* was higher compared to the results of a couple of studies done earlier in Dire Dawa (15.89%) [48] and Jimma (26.6%) in Ethiopia itself [46].

Entero-toxigenic *S. aureus* (load >$10^5$ CFU/g) is one of the most harmful foodborne pathogens found worldwide and the intoxication could be due to improper handling of food including meat [2]. In the current study, its overall isolation rate is 59%, ie., 55.6, 60 and 73.7% were detected in beef, mutton and chevon samples respectively. The rates of isolation from beef samples were comparable to the results of a study done in Jijiga, Ethiopia (52%) [49]; however, this is much higher than that found in a previous study reported from Jimma, Ethiopia [46]. The extent of the isolation of *S. aureus* from chevon samples (73.7%) was more than that found in a study conducted in Jijiga, Ethiopia (47.7%) [49]. A probable reason for this could be the variations in the hygiene being practised in standard abattoirs in comparison to municipal abattoirs. It was also been observed that the majority of butcheries in restaurants, hotels and also in abattoirs did not frequently use disinfectants to clean the contact surfaces, butchery premises, counters and equipment, which played a major role in spoiling hygiene. Consequently, the risk of contamination of meats with pathogens is likely and may raise adverse public health concerns. Also, the mincing of meat could enhance the chances of surface contamination [50]. Besides, sample size, design of the study, methodology used, type of meat samples, as well as geographical location also might have contributed to fluctuations in the rate of isolation of bacteria. A previous study done in the title town reported that the isolation rate of *S. aureus* among food handlers was 7.1% [51]. Detection of these pathogens in meat samples can elevate the chances of diarrheal diseases or even cause an outbreak. Risk factors associated with the contamination of meat samples were not identified exactly in our study, but maybe from the environment, slaughtered animals and handlers [50]. Furthermore, extensive studies are also required to correlate the linkage between contaminated meat and diarrheal diseases in the study area.

Antimicrobial susceptibility profiles observed in the present study revealed that the majority of isolates of *Salmonella* spp. were found to be susceptible to co-trimoxazole (92%) and meropenem (98.6%). A similar trend of susceptibility was observed in a couple of studies done in another part of Ethiopia [52, 53].

We have obtained only a lower level of resistance against ampicillin (17.8%) and this contrast with the results (65% of isolates showed resistance) of a recent study done in Hawassa [52]. In addition, a meta-analysis performed in the country found a higher pooled resistance level of Salmonella isolates in human stools and food of animal origin, which corresponded to 80.6% (95% CI 72.6, 86.7) for ampicillin [54]. Majority of *Campylobacter* spp. isolates were found to be susceptible to ciprofloxacin (95%), followed by co-trimoxazole (89%) and erythromycin (87%). In contrast, a higher level of resistance (95%) to ciprofloxacin was observed in a study conducted on food products of animal origin in Korea [55]. Our results are in agreement with, the pooled antimicrobial resistance rate of animal-derived *Campylobacter* sp. to ciprofloxacin in Ethiopia was 71.2% [45]. Also in the present study, more than 50% of the isolates of *Campylobacter* spp. were found to be resistant to augmentin and penicillin. A similar trend of resistance was observed in a study reported from Spain too [56].

It is important to note that, invariably all the isolates of *Shigella* spp. were susceptible to co-trimoxazole and meropenem. The results also revealed that the majority of isolates of *Shigella* spp. were susceptible to ciprofloxacin (90%) and tetracycline (82%) and this was in contrast with the results of a study done in Nigeria which reported that all the isolates showed a higher degree of resistance to co-trimoxazole and ciprofloxacin [57]. Besides, 90% of these isolates were found to be resistant to ampicillin and this is comparable to the results of a study conducted in another region of Ethiopia (90.6%) [47].

Threescore of the isolates of *E. coli* showed resistance to ampicillin (60%). This was in agreement with the results of a previous study done in Ghana [58] and might be attributed to the continuous application of penicillin derivatives in animals reared for slaughtering [59]. Interestingly, isolates of *E. coli* showed higher levels of susceptibility to ciprofloxacin, co-trimoxazole, chloramphenicol and gentamicin (97–100%) which were similar to the findings of a study done again in Ghana (ciprofloxacin (95.56%), co-trimoxazole (82.22%), and gentamicin (75.56%)) [60]. In contrast, the results of a study done in Dire Dawa, Ethiopia showed that isolates of *E. coli* were highly resistant to all the antimicrobials tested except, tetracycline [48]. These variations observed in the antimicrobial susceptibility profiles could be due to the inconsistencies or non-uniformity in antibiotic prescription policy, usage of antibiotics as veterinary medicines and also because of the blending of antibiotics in animal feed forages.

Notably, 82% of isolates were found to be MRSA and it is slightly lower than the results of a study reported from Jimma, Ethiopia (90%) [46], on the other hand, much higher than the values observed in a study from Colombia (7.5%) [61]. The presence of MRSA isolates in meat samples hints at a fast-growing and risky situation directly affecting the public health system and the community [10]. Livestock-associated MRSA is another risk and there exists a greater chance of a linkage between food animals and human MRSA colonization [62].

The most alarming thing in the current study is that all the isolates were found to be at least resistant to one of the antimicrobials tested. It was found that more than 60% of the isolates were MDR which was higher than that reported in a study conducted in another city of (Hawassa, 36.5%) Ethiopia, indicating that multidrug resistance differs significantly among different regions [52]. In our study, the highest rate of isolation was observed in the case of chevon followed by mutton. However, based on the current set of results, the root cause of higher rates of MDR among the isolates from chevon could not be identified and the exact reason needs to be elucidated by means of future in-depth studies.

Antibiotics are used in food animals for therapeutics and non-therapeutics purposes and such consistent usage can be a major determinant for the emergence of resistant bacteria in meat and can spread to humans [63]. Consuming highly patronized meat types are riskier in this regard. For instance, resistance in Enterobacteriaceae to commonly used antibiotics is widespread as per a recent study conducted in Ghana [58]. Selective pressure due to antibiotic usage in primary production is considered a major source of antibiotic-resistant bacteria in livestock products which parallel with sanitary conditions at slaughter, sail and processing points. All these can affect the profile and intensity of spread along the food chain [63]. Our results revealed the existence of MDR among the isolates of *Salmonella* spp., *Campylobacter* spp., *Shigella* spp., *S. aureus* and *E. coli*. The percentage of MDR isolates of *Salmonella* spp. observed currently was lower than that detected in a previous study done in Bangladesh (89.1%) [59]. The percentage of MDR isolates of *Campylobacter* spp., found in the present study (56.7%) was higher than that previously reported from other regions of Ethiopia (20% and 14.5%) [41, 64], but was much lower than what resulted from a couple of studies done in Brazil (62.8%) [65] and Korea (93.4%) [55]. In the case of *S. aureus*, 59.8% were MDR and this was much higher than that observed in a previous study reported from Addis Ababa, Ethiopia (34.4%) [66]. Multidrug resistance shown by MRSA is currently considered a global threat by WHO. In our study, all the isolates of MRSA were found to be MDR. A recently published excellent review has described the association between the usage of antimicrobial in food animals and the impact of transmission of antimicrobial resistance on humans [63].

Shortcomings of the study include the usage of conventional culture methods and identification of diarrheagenic strains of *E. coli*, enterotoxigenic strains of *S. aureus* and speciation of *Salmonella*, *Shigella* and *Campylobacter* were not done due to the lack of chemicals. Molecular detection of virulence and antimicrobial resistant genes of the major isolates was not performed due to the lack of infrastructure/ facilities.

## Conclusions

This study provided insights into the bacteriological quality of RTE meat and pathogenic bacterial isolates from beef, chevon and mutton which are being served in different hotels and restaurants in Arba Minch town. Overall, the results of our study implied that some of the raw minced meats supplied in all the selected hotels and restaurants in the locality contain higher bacterial loads, which exceed the permissible level. The highlight of this study is the detection of *E. coli*, *S.aureus*, *Salmonella* spp., *Shigella* spp., and *Campylobacter* spp. that pose serious risks to the health of consumers. A higher percentage of multidrug-resistant isolates were also detected, which may result in serious risk of transmission to handlers, consumers and the environment.

## Acknowledgments

We greatly acknowledge the continuous support and encouragement given by both the Department of Medical Laboratory Sciences, College of Medicine and Health Sciences, Arba Minch University and Health and Demographic Surveillances research directorate, Arba Minch University. Thanks are extended to eminent Prof. Dr. K.R. Sabu for the immense help rendered for the English corrections.

## Author Contributions

**Conceptualization:** Tomas Tonjo, Aseer Manilal, Mohammed Seid.

**Data curation:** Tomas Tonjo, Aseer Manilal, Mohammed Seid.

**Formal analysis:** Tomas Tonjo, Aseer Manilal, Mohammed Seid.

**Investigation:** Tomas Tonjo, Aseer Manilal, Mohammed Seid.

**Methodology:** Tomas Tonjo, Aseer Manilal, Mohammed Seid.

**Project administration:** Tomas Tonjo.

**Resources:** Tomas Tonjo, Aseer Manilal, Mohammed Seid.

**Software:** Tomas Tonjo, Aseer Manilal, Mohammed Seid.

**Supervision:** Aseer Manilal, Mohammed Seid.

**Validation:** Tomas Tonjo, Aseer Manilal, Mohammed Seid.

**Visualization:** Tomas Tonjo, Aseer Manilal, Mohammed Seid.

**Writing – original draft:** Tomas Tonjo, Aseer Manilal, Mohammed Seid.

**Writing – review & editing:** Aseer Manilal, Mohammed Seid.

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
