## [Decision Letter · Decision Letter 0]

12 Jul 2022

PONE-D-22-11268Bacteriological quality and antimicrobial susceptibility patterns of isolates of ready-to-eat raw minced meat from hotels and restaurants in Arba Minch, EthiopiaPLOS ONE

Dear Dr. Manilal,

Thank you for submitting your manuscript to PLOS ONE. After careful consideration, we feel that it has merit but does not fully meet PLOS ONE’s publication criteria as it currently stands. Therefore, we invite you to submit a revised version of the manuscript that addresses the points raised during the review process.

We look forward to receiving your revised manuscript.

Kind regards,

Jasbir Singh Bedi

Academic Editor

PLOS ONE

Journal Requirements:

Reviewers' comments:

Reviewer's Responses to Questions

**Comments to the Author**

1. Is the manuscript technically sound, and do the data support the conclusions?

Reviewer #1: Partly

2. Has the statistical analysis been performed appropriately and rigorously? 

Reviewer #1: No

3. Have the authors made all data underlying the findings in their manuscript fully available?

Reviewer #1: Yes

4. Is the manuscript presented in an intelligible fashion and written in standard English?

Reviewer #1: No

5. Review Comments to the Author

Reviewer #1: The manuscript PONE-D-22-11268 entitled “Bacteriological quality and antimicrobial susceptibility patterns of isolates of ready-to eat raw minced meat from hotels and restaurants in Arba Minch, Ethiopia” provides insights into the bacteriological quality of ready-to-eat raw minced meat in Arba Minch in Eithopia. It is worth mentioning that this type of study is important in places where people consume raw meat also as is the case in cuurent scenario. The study is all the more important as many isolates were multi drug resistance which is a grim situation. However, the manuscript has some major lacunae and shortcomings, which should be considered before its approval.

Confirmation of isolates by biochemical characterization and molecular tests is very important, including identification of pathogenic isolates of the bacteria.

The discussion is not adequately written.

6. PLOS authors have the option to publish the peer review history of their article (what does this mean?). If published, this will include your full peer review and any attached files.

Reviewer #1: No

---

## [Author Response · Author response to Decision Letter 0]

29 Jul 2022

Response to the reviewer

Comment

The manuscript PONE-D-22-11268 entitled “Bacteriological quality and antimicrobial susceptibility patterns of isolates of ready-to eat raw minced meat from hotels and restaurants in Arba Minch, Ethiopia” provides insights into the bacteriological quality of ready-to-eat raw minced meat in Arba Minch in Eithopia. It is worth mentioning that this type of study is important in places where people consume raw meat also as is the case in cuurent scenario. The study is all the more important as many isolates were multi drug resistance which is a grim situation. However, the manuscript has some major lacunae and shortcomings, which should be considered before its approval.

Response- Thanks for your encouraging Comments. 

We have revised the manuscript substantially as per your Comments and suggestions

Comment

Abstract: Make it more concise, staring from a cross sectional study…Include lines on MDR and MRSA also.

Response-Thanks for your valuable Comment. The abstract part is substantially revised and included a line on MDR and MRSA as per your suggestion.

Comment

INTRODUCTION 

The Introduction includes many grammatical errors. The following sentences need to re-written as below.

Response- Thanks for your valid support. The introduction part is revised as per your direction.

Comment

Line 57: Ethiopia ranks second after Nigeria in this regard and FBDs pose a serious threat to the health of the people in the country.

Response- FBDs is included

Comment

Line 67: The most common source of the bacterial causes of diarrheal diseases via meat chains is food of animal origin, environment, meat handlers/processors and processing equipment.

Response- Corrected in the revised manuscript

Comment

Line 69: Even though the source of the bacteria varies, raw meat is one of food of animal origin which has been confirmed as one of the common vehicles of foodborne diseases.

Response- Corrected in the revised manuscript

Comment

Line 72: Moreover, The most common food safety problems …….

Response-Corrected

Comment 

Line 78: Transmission of these resistant bacteria to humans via meat is most evident ………

Response-Corrected

Comment

Line 84: The problem is even more alarming in developing countries, where there is an enormous burden of infectious diseases, accompanied by lack of surveillance networks, paucity of testing laboratories, and inadequate diagnostics.

Response- Corrected

Comment

Line 87: Although healthy meat from beef cattle, sheep and goat …

Response- Corrected

Comment

Line 91: The general masses comprising all age groups of the study area, Arba Minch town….

Response- Corrected

Comment

Line 93: Nevertheless, many hotels and restaurants are also serving ready-to-eat raw meats, without assessing its bacteriological quality, which is further aggravated by lack food safety and inspection department in the town. 

Response-Corrected

Comment

Line 95: Literature survey indicates that studies in this context are sorely lacking.

Response -Corrected

Comment 

MATERIALS AND METHODS 

The study was carried out for a period of six months. However, the samples were collected for only 2.5 months and that too has been wrongly mentioned as 01 July to 31 Oct (Line 116), which comes out to be four months. Kindly rectify it and remain consistent throughout the text and tables for the period of study.

Response- Corrected as 6 months 01 July to 31 December throughout the manuscript to keep the consistency 

Comment

Line 105: The number of animals slaughtered per day varied from time to time, for instance, during fasting, festival and non-fasting periods. 

Response - Corrected

Comment

Line 106: The average number of cattle slaughtered per month approximates to 600 excluding sheep and goats. 

Response-Corrected

Comment

Line 109: Ethical approval for this study was obtained from the Institutional Review Board of College of Medicine and Health Science, Arba Minch, Eithopia (Ref. No IRB/174/12/17/03/2020).

Response- Corrected

Comment

Line 117: A total of 260 samples were randomly collected comprising 170 beef, 50 sheep and 40 goat meat samples. 

Response- Corrected

Comment

Line 118: One beef sample and two samples from goat meat……………..

Response-Corrected

Comment

Line 122: Rewrite

Response-Corrected

Comment

Line 125: Inclusion criterion was RTE raw minced meat (regional name of RTE: Kurt/Kitfo) that is processed for direct consumption, whereas the exclusion criterion was minced meat that is processed for cooking.

Response -Corrected

Comment 

Line 130: Collected samples were drawn (from where…………..??????) by sterile tweezers and placed in a sterile plastic bag and immediately transported in ice box to the Medical Microbiology and Parasitology Laboratory, Department of Medical Laboratory Science for bacteriological analysis, within 2 h of collection. 

Response-Corrected

Comment

Line 141: After incubation, plates showing between 30 and 300 colonies were counted and the results were calculated according to ISO 7218:2007 using the following formula [15] This ISO standard has been revised and the annex D pertains to Salmonella. The reference and the formula needs to be cross checked.

Response- Reference and formula is cross-checked and updated reference has been given in the revised manuscript

Comment

Line 153: The pdf file of reference 16 to be furnished please. Moreover, it’s a very old reference, maybe substituted with latest available.

Response-Corrected

Comment

Line 155: About 0.1ml from each dilution was spread on to the such as MaConkey agar and mannitol salt agar (Himedia, India) for the total coliform count and S. aureus count respectively and incubated at 37°C for 24 hours.

Response-Corrected

Comment

Line 159 and 164: use were instead of was

Response-Corrected

Comment 

Line 171: S. aureus in italics

Response-Corrected

Comment

Line 183: Included

Response-Corrected

Comment

Line 196-197: Salmonella Typhimurium

Response-Corrected

Comment 

Overall the data is poorly presented in the results. 

Response-Thanks for your valuable Comment

Comment

Line 215: On the contrary, respectively, 13.2% (n=34), 17.5% (n=45) and 17.5% (n=45) of meat samples had unacceptable ranges of bacteriological load with respect to TVC, TCC and TSC, respectively. The word respectively has been wrongly written in the middle of the sentences at many places throughout the text. It should be mentioned at the end of the sentence (where ever required) after putting a comma. Should be changed and corrected throughout.

Response- The word respectively has been shifted to the end of the line where and when required.

Comment

Line 218: The most alarming factor is that, 10.5% (n=4), 8% (n=4) and 1.8% (n=3) of beef, sheep and goat meat 219 samples respectively were contaminated with S. aureus and their TSC loads remained in a 220 potentially harmful range (Table 1)…. These figures donot match with the figures quoted against different species in Table No. 1

Response- Corrected in the revised manuscript

Comment-

Line 221: were inplace of are and was inplace of is… correct the entire text for the grammatical errors..It has to be presented in past tense rather than present.

Response- Corrected

Comment

Line 223: use permissible instead of permitted

Response- Corrected

Comment

Line 245: six

Response- corrected 

Comment

Line 247: correct spellings of susceptibility 

Response – Corrected 

Comment

Line 258: In case of E. coli isolates, a higher degree……………

Response- corrected 

Comment

Line 269: 51%

Response-corrected 

Comment

Try replacing sheep and goat meats by mutton and chevon, respectively in text.

Response-Replaced in the revised manuscript where and when required

Comment

Table No.2 How can the TCC and TSC values for beef be lesser than sheep and goat meats, when the TVC is on the higher side?

Response-Replaced in the revised manuscript where and when required

Comment

Confirmation of isolates by biochemical characterization and molecular tests is very important, including identification of pathogenic isolates of the bacteria.

Response- Thanks for your invaluable Comment. However, due to the lack of infrastructure and facility, we are unable to perform molecular detection of genes. This is included under the limitation as per your suggestions.

Comment

DISCUSSION

The discussion is not adequately written. The first page is a repetition of the results only. Simply comparisons have been made with other references in Eithopia. Only one global reference is there, which is Nepal. Comparison of the finding in meat may be done with that of human and animal prevalence too. Risk factors maybe identified to justify the statements made. The discussion on AMR and MRSA should be more focused.

Response- Thanks for your Comment. We have extensively modified the discussion part by including relevant references, ie. 10, where and when required. Risk factors are included in the discussion. AMR and MRSA discussed in the revised manuscript.

---

## [Editor Report · Decision Letter 1]

16 Aug 2022

Bacteriological quality and antimicrobial susceptibility profiles of isolates of ready-to-eat raw minced meat from hotels and restaurants in Arba Minch, Ethiopia

PONE-D-22-11268R1

Dear Dr. Asser

We’re pleased to inform you that your manuscript has been judged scientifically suitable for publication and will be formally accepted for publication once it meets all outstanding technical requirements.

Kind regards,

Jasbir Singh Bedi

Academic Editor

PLOS ONE
---

## [Editor Report · Acceptance letter]

24 Aug 2022

PONE-D-22-11268R1 

Bacteriological quality and antimicrobial susceptibility profiles of isolates of ready-to-eat raw minced meat from hotels and restaurants in Arba Minch, Ethiopia 

Dear Dr. Manilal:

I'm pleased to inform you that your manuscript has been deemed suitable for publication in PLOS ONE. Congratulations! Your manuscript is now with our production department. 

Kind regards, 

on behalf of

Dr. Jasbir Singh Bedi 

Academic Editor

PLOS ONE